# The Laboratory Diagnosis of *Neisseria gonorrhoeae*: Current Testing and Future Demands

**DOI:** 10.3390/pathogens9020091

**Published:** 2020-01-31

**Authors:** Thomas Meyer, Susanne Buder

**Affiliations:** 1Department of Dermatology, Venerology and Allergology, St. Josef Hospital, Ruhr-University, 44791 Bochum, Germany; 2German Consiliary Laboratory for Gonococci, Department of Dermatology and Venerology, Vivantes Hospital Berlin, 12351 Berlin, Germany; susanne.buder@vivantes.de

**Keywords:** gonorrhea, diagnostic, microscopy, culture, antimicrobial resistance, NAAT, point-of-care test, microfluidic

## Abstract

The ideal laboratory test to detect *Neisseria gonorrhoeae* (*Ng*) should be sensitive, specific, easy to use, rapid, and affordable and should provide information about susceptibility to antimicrobial drugs. Currently, such a test is not available and presumably will not be in the near future. Thus, diagnosis of gonococcal infections presently includes application of different techniques to address these requirements. Microscopy may produce rapid results but lacks sensitivity in many cases (except symptomatic urogenital infections in males). Highest sensitivity to detect *Ng* was shown for nucleic acid amplification technologies (NAATs), which, however, are less specific than culture. In addition, comprehensive analysis of antibiotic resistance is accomplished only by in vitro antimicrobial susceptibility testing of cultured isolates. As a light at the end of the tunnel, new developments of molecular techniques and microfluidic systems represent promising opportunities to design point-of-care tests for rapid detection of *Ng* with high sensitivity and specificity, and there is reason to hope that such tests may also provide antimicrobial resistance data in the future.

## 1. Introduction

*Neisseria gonorrhoeae* (*Ng*) infections belong to the most frequent sexually transmitted infections with worldwide around 87 million new infections per year according to WHO estimations [1]. Of these, about 4 million occur in Europe, North America, Australia, and New Zeeland. The vast majority (>80 million) of gonococcal infections are in low- and middle-income countries of Asia, Africa, Latin America, and the Caribbean [1], but increasing incidence is reported in Europe and the USA [2,3]. In Europe, men having sex with men (MSM) accounted for almost half of the reported cases [3]. Transmission of the bacteria is by direct mucosal contact and may lead to infections at the urethra, endocervix, rectum, pharynx, or conjunctiva. While 90% of male urethral infections present with discharge or dysuria, less than 50% of female urethral and cervical infections are symptomatic, and most rectal infections and almost all pharyngeal infections are asymptomatic [4,5,6]. By transluminal dissemination, starting from the urethral or endocervical mucosa, *Ng* may cause ascending infections resulting in epididymo-orchitis, salpingitis, and pelvic inflammatory disease (PID) [7,8,9]. In rare cases, the bacteria may spread systemically resulting in severe complications like fever/septicemia, arthritis, tenosynovitis, endocarditis, or vasculitis [10]. In addition, gonococcal infection in pregnancy is associated with adverse pregnancy outcomes, like low birth weight infants, small for gestational age infants, and transmission to newborns that may result in conjunctivitis (ophthalmia neonatorum) and oropharyngeal infections [11,12].

Due to the variety of symptoms that are largely not specific for gonorrhea, timely and accurate laboratory testing of symptomatic patients is required including resistance testing of *Ng*-positive cases for targeted antimicrobial treatment. In addition, screening of key populations to identify and treat asymptomatic infections is no less important to reduce transmission of infection and to control spreading of antibiotic resistance. Hence, the following conditions indicate laboratory testing of gonococcal infection: diagnostic evaluation of clinical symptoms, treatment monitoring, sexual partner(s) diagnosed with *Ng*, other diagnosed STI, new sexual partners or frequently changed sexual partners, and sexual abuse. In the absence of laboratory diagnostics in resource-poor settings, *Ng* infection is usually identified by clinical manifestations combined with medical history and a typical incubation period (2–8 days). However, even classical clinical symptoms like male purulent discharge and urethritis or purulent vaginal discharge or proctitis are not sufficient evidence of gonorrhea, as various other pathogens can cause very similar or identical images. The syndromic approach may possibly suffice for urethral discharge in men but has poor sensitivity and specificity to detect infections in women and non-urethral infection in men, potentially resulting in inadequate treatment with the risk of inducing resistance. It is therefore essential to make laboratory testing available to resource-limited settings.

Laboratory diagnosis of gonococcal infection is established by direct detection of the pathogen in urogenital, anorectal, pharyngeal, or conjunctival swab specimens or first-catch urine. Presently, several different techniques are available for *Ng* detection, of which culture and nucleic acid amplification technologies (NAATs) are best suited [13]. Microscopy of stained urogenital specimens can also be used in certain cases. DNA probe assays, antigen tests, and serology to detect antibodies against *Ng* are not recommended for laboratory testing due to insufficient sensitivity and specificity [13]. 

During the last decades, diagnostic procedures have been improved continuously, resulting in a better management of individual patients. There are, however, some public health issues to be considered in this context.
Improvements of *Ng* testing resulted in increased detection rates that may have influenced epidemiologic data (i.e., higher detection rates do not necessarily indicate an increase in transmitted infections but may just reflect more sensitive and more frequent testing). For instance, introduction of NAATs in routine diagnostic testing have shown pharyngeal and rectal infection to be much more prevalent than previously assumed [14].Since rectal and pharyngeal infections, as well as cervical infections in women, are frequently asymptomatic and will be missed by symptom-based examinations, laboratory testing should consider inclusion of both urogenital, anorectal, and pharyngeal samples, depending on sexual behavior, to identify infected individuals with higher sensitivity [5,15,16].NAAT-based treatment monitoring has improved identification of treatment failures that particularly relate to pharyngeal infections [17]. Considering the presence of non-gonococcal *Neisseria* species at the pharyngeal mucosa that may transfer resistance to *Ng* [18,19], the pharynx has been suggested an important site for resistance development. Currently, the frequency and impact of genetic exchange in the pharynx is not known exactly but is important to be clarified, as it would strongly support pharyngeal screening and clearance of pharyngeal infections to be essential.

The objective of this review article is to summarize current diagnostic procedures for *Ng* detection according to recommendations of several guidelines and to review recent advances and novel developments that may potentially improve *Ng* diagnostic testing, based on publications primarily of the last 5 years identified by a PubMed literature search.

## 2. Microscopy

Direct microscopy is suitable in defined settings for the detection of *Ng* as a point-of-care test. Depending on the clinical picture, direct microscopy may be a valid diagnostic tool in settings with more modest resources.

For direct microscopy, two different staining methods are used: methylene blue staining and Gram´s staining. To prepare a staining preparation, the secretion is spread out in a thin layer on a microscope slide and is heated for fixation. For methylene blue staining, the slide is coated with 1% aqueous methylene blue solution or immersed in a cuvette. After a short exposure (15 s), the preparation is rinsed with water and dried between groundwood filter paper. At methylene blue staining, all bacteria turn blue. It should only be used as a diagnostic criterion for uncomplicated male urethritis in combination with typical clinical symptoms. In women’s gonorrhea and all other manifestations of disease, Gram´s staining is required [20,21]. Complete staining sets are commercially available. It allows the differentiation into Gram-negative (red) and Gram-positive (blue-violet) bacteria following a stepwise procedure of staining.

At high magnification (1000×), the leukocyte-rich sites are searched and examined with oil immersion. The typical pattern for gonococci is the paired (diplococci), piled, intraleukocytic storage. Diplococci are perpendicular to each other, have the same size and bean or kidney shape. In Gram´s staining, *Ng* shows as Gram-negative diplococci, often intracellular in polymorphonuclear leukocytes with typical morphology (Figure 1). However, detection of extracellular diplococci, especially in connection with a typical clinical picture, also indicates the presence of *Ng*. The detection is doubtful if atypical Gram-negative or Gram-labile diplococci are present. In symptomatic male urethritis, the sensitivity of Gram´s staining is up to 95% and highly specific (97%) for an experienced examiner. In endocervical samples, the sensitivity decreases to 40–60% [21,22]. In asymptomatic patients and in pharyngeal and rectal smears, the sensitivity is extremely low, and the method is not recommended [23]. Other bacteria, especially other *Neisseria* species, which have a similar morphology, compromise the microscopic result in extragenital specimens. In asymptomatic patients, the load of gonococci to be detected is usually too low.

## 3. Culture

Antimicrobial resistance in *Ng* is a severe problem worldwide, and reliable results are indispensable preconditions for a successful therapy. Bacterial culture is sensitive and highly specific. In urogenital specimens, sensitivity may reach 85–95% under optimal conditions [23,24], and specificity is up to 100% when species identification is performed, as shown below. Up to now, it is the only method which allows complete antimicrobial susceptibility testing.

As an example, culture of *Ng* from a male urethral swab is shown in Figure 2. Cultivation does not succeed equally well from every sample material. Smear materials from the urethra and cervix are favorable. Bacterial culture from conjunctival, rectal, and oropharyngeal samples require optimal growth conditions, are time consuming, and often frustrating, especially in the case of throat swabs. Vaginal swab specimens and urine are rarely successfully cultivated [23,25,26].

Gonococci are very demanding and fastidious pathogens. They do not tolerate dehydration and should be inoculated immediately after swab collection onto culture media (nutritious selective culture medium and non-selective culture medium) [27]. Culture plates must be incubated at 35–37 °C and high humidity (70–80%), pH 6.75–7.5, and in a 4–6% CO_2_-enriched atmosphere. After 18–24 (–48) h, small, shiny gray colonies appear, whereby colony growth variations are possible [23]. After cultivation, identification of *Ng* is assessed by combining several detection methods. A presumptive identification is performed by microscopic Gram´s staining preparation and positive cytochrome oxidase reaction. To confirm the identification and distinguish between other *Neisseria* species like *Neisseria meningitidis* and apathogenic *Neisseria* spp. especially in extragenital sites, biochemical tests, immunological test, spectrometric test, or molecular test are applied. In biochemical identification tests (e.g., API-NH, bioMerieux), various enzymatic reactions and metabolic reactions are displayed by color change. The determined numerical profile allows the identification of the pathogen [28]. Alternative coagglutination tests with gonococcal-specific monoclonal antibodies (e.g., Phadebact Monoclonal GC test, MKL diagnostics; Gonocheck II TCS Biosciences Ltd.) can be performed [28,29].

Furthermore, mass spectrometric identification of *Ng* as a culture-based detection method can be used. MALDI-TOF (matrix-assisted laser desorption ionization time-of-flight) mass spectrometry identifies bacteria that can be taken directly from the agar plate. The result is a spectral fingerprint that can be assigned to the respective microorganism. The method has become established for the detection of *Ng* [30], with a reported positive predictive value of 99.3% [31,32]. However, results should be interpreted with caution for *Ng* and commensal *Neisseria* species when isolated from extragenital and oropharyngeal samples [33]. Molecular confirmation of *Ng* can also be performed using NAATs (see Section 5). Gonococcal typing and genome sequencing are mostly reserved for scientific, epidemiological, and forensic questions.

## 4. Antimicrobial Susceptibility Testing

Antimicrobial susceptibility testing is one of the most important procedures while processing *Ng*. This allows a reliable statement about the possible effectiveness of an antimicrobial therapeutic agent. Usually, the testing is performed as an indication of the minimum inhibitory concentration (MIC) of an antimicrobial agent in μg/ml or mg/l that inhibits growth. Various breakpoint standards are available for the assessment of susceptibility. It should therefore always be clear according to which standard the assessment is made. Currently, the following two standards are mainly used: CLSI (Clinical and Laboratory Standards Institute) (https://clsi.org/) or EUCAST (European Committee for Antimicrobial Susceptibility Testing) (http://www.eucast.org/clinical_breakpoints/). In addition, national antibiotic sensitivity test committees exist and should be consulted. MIC breakpoints were divided into three categories: “susceptible”, “intermediate”, and “resistant”. Since 2019, EUCAST has been introducing a new classification split into “susceptible“, “susceptible with increased exposure“, and “resistant“. In comparison, the breakpoints for *Ng* of the annually revised and freely accessible recommendations of EUCAST are slightly lower than the ones recommended by CLSI. EUCAST provides no information to zone diameter breakpoints and hence no information on disc diffusion testing. Currently, the main difference for *Ng* breakpoints between the two standards exists for azithromycin. CLSI provides no breakpoints for azithromycin. EUCAST changed from defined breakpoints to an epidemiological cut-off (ECOFF) in 2019. The ECOFF of 1 mg/L applies now to report acquired resistance.

Before performing susceptibility testing, it is important to select a test panel that is appropriate for the pathogen, the expected resistance, and the possible therapy options. There are three main test options for determining susceptibility: agar dilution method, MIC gradient strip test method, and disc diffusions assay. Regardless of the test method, strict and constant quality assurance, the use of WHO control strains, and intra-laboratory and external quality control assessments are required [34,35].

A limited qualitative estimation of antimicrobial susceptibility can be obtained by using a disc diffusion assay [36]. Thereby, discs containing defined antibiotic concentrations are placed on the agar surface. The antibiotic agent diffuses into the culture medium and inhibits growth. After incubation, the inhibition zone diameters are measured in mm. The growth inhibition zone is considered an approximation of the susceptibility (Figure 3). Several disc diffusion methods are described, and the method is only recommended when MIC determination cannot be performed, e.g., due to limited resources [37,38,39]. Therapeutically relevant disc diffusion results should be supplemented by confirmation tests using other methods.

The agar dilution method is the WHO-recommended gold standard method for antimicrobial susceptibility testing in *Ng*. A defined series of concentrations of an antibiotic substance is incorporated into nutrient medium. The corresponding growth inhibition is read off in relation to the rising concentration of the antibiotic. The lowest concentration that inhibits growth gives the MIC value in μg/ml (mg/l). However, this method is complex and predominantly suitable for a large number of tests [23]. A similar approach with broth microdilution has been developed but not established as a standard testing method so far [40].

Therefore, the standardized and quality-assured MIC gradient stripe test method (Etest), which correlates with the agar dilution method, is currently the preferred method [23,41,42]. MIC gradient stripe tests are plastic test strips with a predefined concentration gradient for a single antibiotic (Figure 4). The antibiotic agent diffuses into the culture medium, and the elliptical growth inhibition can be read off after incubation by using the printed MIC scale in μg/mL (mg/L). In 2018, various manufacturers of MIC strip test (MIC Test Strips—Liofilchem, M.I.C.Evaluator—Oxoid, and Ezy MIC Strip—HiMedia) were compared and evaluated compared to the reference Etest by bioMérieux for *Ng* testing. None of the tests met the high standards of Etest. M.I.C.Evaluator strips did not offer the antibiotic panel relevant for *Ng* testing. Ezy MIC strips showed inconsistent accuracy and quality. Liofilchem MIC Test Strips proofed relatively accurately but were still not fully comparable to Etest (bioMerieux) results [42,43]. In particular, testing of azithomycin is difficult, and test results can vary considerably [44].

Some *Ng* strains carry a plasmid that encodes a high-level resistance to penicillin. This can be rapidly detected by color change using the Nitrocefin test. However, a negative ß-lactamase test does not exclude low-level penicillin resistance [45].

Recent studies show that early transcriptional changes upon exposure to antibiotics may be used to assess antimicrobial resistance (AMR). For *Ng*, transcripts have been identified that are differentially regulated in strains susceptible or resistant to ceftriaxone, azithromycin, or ciprofloxacin [46,47,48]. Quantification of these antibiotic-responsive transcripts results in a signature indicating susceptibility or resistance that may be used as a diagnostic tool in the future. In contrast to DNA-based resistance, test levels of antibiotic-responsive transcripts are independent of genetic mechanisms of resistance, but variance of gene expression may also result from genetic distance effects. Expression levels of porB and rpmB transcripts have been described to diagnose ciprofloxacin resistance in *Ng* isolates [46], but when testing a genetically more diverse panel of isolates’ expression levels of the two markers, they were no longer able to differentiate between susceptible and resistant strains [47]. The technique has great diagnostic potential, but large numbers of diverse *Ng* isolates from all over the world need to be tested to confirm that specific transcript levels indicate susceptibility or resistance to specific drugs.

## 5. NAATs

NAATs are the most sensitive techniques to detect *Ng*. Sensitivity and specificity of *Ng* NAATs is generally >95% and >99% in swabs and male first-catch urine (FCU) [4,13]. Currently available commercial tests are based on polymerase chain reaction (PCR) or isothermal transcription-mediated amplification. A list of FDA-approved commercial NAATs as of December 2019 is shown in Table 1. Superiority of NAATs over culture has been demonstrated in a number of studies [13,49,50,51,52,53,54]. The higher sensitivity of NAAT is partly due to the independence of viable bacteria and applies especially to extragenital specimens [13,14]. Another advantage compared to culture relates to the utility of diverse specimen types that are easier to handle, as no viable bacteria are required [50]. Moreover, NAATs are easier to perform and faster than culture with less hands-on time and the capability of automation allowing high throughput testing [55,56,57]. In addition, many commercial NAATs were designed to detect both *Ng* and *Chlamydia trachomatis* (*Ct*) in a single reaction [13].

Although sensitivity of NAATs is superior to other detection methods, it should be considered that diagnostic accuracy may be affected by genetic variations and the genomic plasticity of *Neisseria*. Loss or modification of target regions were shown to reduce sensitivity [58,59], whereas specificity may be diminished by cross-reactive non-pathogenic *Neisseria* species as well as horizontal transfer of *Ng* gene sequences to commensal *Neisseria* [60,61]. Another disadvantage of NAAT-based detection is the lack of information about AMR that still requires isolation of the bacteria by culture and subsequent susceptibility testing. Many commercial NAATs use specific specimen collection kits inappropriate for bacterial culture. However, NAATs may also work properly with nylon flocked swabs in ESwab collection kits [62], from which *Ng* culture succeeded in up to 70% after storage at 4 °C for one day [63], allowing a deferred culture strategy depending on antecedent NAAT results.

To detect male urethral gonococcal infection, first-catch urine (FCU, the first 10–20 mL of micturition) and urethral swabs are equally good, whereas in females, vaginal or endocervical swabs are more sensitive than FCU [13]. FCU is typically self-collected, but vaginal swabs and even meatal swabs may also be collected by the patients themselves and were shown to attain reliable NAAT results [64,65,66]. Accordingly, guidelines recommend NAAT testing of FCU in men and self-collected vaginal swabs in women, respectively, for laboratory diagnosis of urethral/cervical *Ng* infection [4,13,67].

NAATs are also the most sensitive tests to diagnose extragenital *Ng* infection [14,68,69] and are therefore recommended for laboratory diagnosis of rectal or pharyngeal infection [13,70]. The relevance of testing extragenital sites is mainly based on two findings: (i) additional testing of pharyngeal and rectal swabs was shown to increase the number of infected individuals [5,16]; (ii) patients can be infected at multiple sites [71], and pharyngeal infections are more difficult to treat [72]. 

Positive NAAT results obtained with extragenital specimens should be confirmed by detection of an alternative target to exclude false positive results due to cross reactivity with commensal *Neisseria* [73]. Confirmatory testing should also be considered in populations with low prevalence (i.e., screening of low-risk populations) with NAAT positive predictive values (PPV) less than 90% [4,74,75]. For routine diagnostic testing, a second test for confirmation seems impractical but may be bypassed using dual target assays including two *Ng* target regions that both need to be amplified for a positive test result [76,77].

Until recently, no NAATs were FDA-cleared for testing extragenital specimens. Thus, compliance with Clinical Laboratory Improvement Amendments (CLIA) for test modifications or equivalent regulatory standards for quality assurance were required. In April 2019, however, two *Ct/Ng* NAATs received FDA approval for testing rectal and pharyngeal specimens (Table 1).

The panel of pathogens included in NAATs for STI testing was recently extended further. Next to *Ct* and *Ng*, *Mycoplsma genitalium* (*Mg*) and *Trichomonas vaginalis* (*Tv*) were incorporated in two commercial assays [57,78]. Performance of BD Max CT/GC/TV assay to detect the three organisms in urine, endocervical, and vaginal swabs from 1990 female and 840 male subjects was consistent with comparator assays for *Ct/Ng* or *Tv* with a sensitivity >95.5% and specificity >98.6% for *Ng* in all specimen types [57]. Similarly, by analyzing 441 urine specimens with the BioRad real-time Dx CT/NG/MG assay, highly concordant results to comparator assays for *Ct/Ng* or *Mg* were described that resulted in a calculated sensitivity and specificity of 92% and 100%, respectively, for *Ng* detection [78].

Some multiplex NAATs allow testing an even more extensive panel of STI pathogens. Selected commercial multiplex NAATs are shown in Table 2. These assays differ with respect to amplification techniques and detection of amplified products, as well as the time to result and the STI panel included. Performance evaluations were published for some of these tests and generally showed good agreement with diagnostic assays for single targets. For instance, the AmpliSense Multiprime FRT real-time PCR assay for *Ct*, *Ng*, *Tv*, and *Mg* was evaluated by comparing it with APTIMA tests for *Ct*, *Ng*, *Tv*, and *Mg* on 209 vaginal swabs and 498 female FCU and 554 male FCU [79]. The authors reported excellent sensitivity and specificity for all pathogens except *Mg*, which lacks sufficient sensitivity. For *Ng*, AmpliSens and APTIMA AC2 results were 100% concordant. The STI FilmArray was applied to 295 clinical specimens and results compared to standard testing [80]. For *Ct* and *Ng*, Roche Amplicor was used as a comparator test and results for *Ct* and *Ng* were 98% and 97% concordant [80]. The STI multiplex assay from Seegene (Anyplex II) uses real-time amplification with multiple primers for seven different STI pathogens. Amplified products were specified by a combination of fluorescence labels and melting point analysis of specially designed oligonucleotides used as hybridization probes. Anyplex II test results largely correspond to results of other diagnostic tests for individual pathogens [81,82]. In a French study, *Ng* test results of 213 specimens obtained with Abbott CT/NG and Anyplex II were 97.2% concordant. Using Abbott CT/NG as a reference test, sensitivity and specificity of Anyplex II were 90% and 98.4%. However, further analysis of samples with discrepant results confirmed one of three Anyplex-positive/Abbott-negative as positive and one of three Anyplex-negative/Abbott-positive test results as negative, indicating sensitivity and specificity of Anyplex II may be even higher [82].

In summary, several studies have shown high clinical sensitivity and specificity of STI multiplex NAATs not inferior to standard NAAT tests for single pathogens or duplex assays. These assays may potentially improve STI diagnostics due the higher content of information. However, depending on the test panel, a controversial discussion about the relevance of test results arose, especially considering microbial agents with low pathogenic potential (like *Ureaplasma parvum* or *Mycoplasma hominis*).

## 6. Rapid Tests and Point-of-Care Tests (POCT)

Although NAATs are considered the primary tests to detect *Ng*, their use in low- and middle-income countries is greatly limited due to relatively high costs. Management of patients in areas with limited access to laboratory testing is based mainly on clinical symptoms (syndromic-guided management) that, however, may result in inappropriate treatment, potentially increasing the risk of AMR development and spreading. For these regions, low-priced rapid diagnostic tests for *Ng* that can be performed independent from expensive laboratory equipment at the point of care are considered an important approach for confirmation of diagnosis and subsequent targeted treatment. There is also a demand for rapid diagnostic tests in countries with well-developed health care systems. Testing in a central laboratory is associated with increased turnaround times due to sample transport and reporting of test results that requires a follow-up visit of patients in order to start treatment in case of positive test results. Rapid diagnostic tests (RDTs) that can be performed at the point of care may produce test results within a timeframe the patient is willing to wait, allowing initiation of antibiotic treatment and instigation of partner notification at the same visit. Thus, early diagnosis and treatment of *Ng* infection by RDTs may potentially reduce ongoing transmission.

In principle, microscopy can be considered an *Ng* RDT that may be performed at the point of care, given the availability of a microscope. However, microscopic evaluation requires skilled investigators and lacks sensitivity in asymptomatic infections as well as in anorectal and pharyngeal specimens. Due to the presence of non-pathogenic *Neisseria*, specificity is also impaired when analyzing rectal or pharyngeal samples [83].

Other *Ng* RDTs are based on antigen-detection by immunochromatography (lateral flow assays) or optical immunoassays. Several immunochromatographic *Ng* RDTs were evaluated in clinical studies and consistently lack sufficient diagnostic performance with a sensitivity between 12.5% and 94% and a specificity between 89% and 99.8% [84,85]. Sensitivity may be even lower, as some studies used culture or an outdated PCR test with suboptimal sensitivity as the reference method [86,87,88]. In addition, the PPV of 97% reported in a Japanese study results from analysis of a specimen collection with >50% gonorrhea prevalence that usually does not reflect real situations. In another study, 100% sensitivity and 93% specificity was reported for the Biostar Optical immunoassay; this study, however, included only 5 *Ng*-positive urine specimens [89]. In conclusion, due to insufficient sensitivity and/or specificity, *Ng* RDTs depending on antigen detection are unsuitable to detect *Ng* infection [84].

In contrast, molecular PCR-based rapid tests were shown to have a much better diagnostic performance, comparable to that of reference laboratory NAATs. The GeneXpert CT/NG assay simultaneously detects *Ng* and *Ct* in a closed system and can be used at the point of care. However, the test does not fulfill the classic ASSURED criteria [90] for POCT (i.e., affordable, sensitive, specific, user-friendly, rapid and robust, equipment-free, and deliverable), as it is high-priced, needs electricity, and takes approximately 90 min. Therefore, the GeneXpert assay is usually referred to as a near-patient test rather than a POCT. By analyzing FCU from males, as well as vaginal and cervical swabs, the assay was able to detect *Ng* and *Ct* with high sensitivity (98–100%) and specificity (99.9–100%) that did not differ from reference laboratory NAATs [91]. The significantly higher clinical sensitivity of Xpert results from the low detection limit of 10 *Ng* genome copies [92], which is much lower than for antigen-based assays. The excellent performance was recently confirmed in another evaluation study on vaginal swabs from young South African women [93]. The high specificity results from amplification of two highly specific chromosomal targets that are both required for a positive test result. Therefore, the assay was also evaluated for anorectal and pharyngeal swabs, usually containing numerous commensal Neisseria species. Whereas no significant differences were found by comparing GeneXpert CT/NG with Aptima Combo2 in self-collected rectal swabs [94], sensitivity of GeneXpert was lower when analyzing male pharyngeal samples [95].

Main disadvantages of the GeneXpert system are the relatively high costs and the test duration of about 90 min that may compromise immediate antibiotic treatment in case of positive results if patients are not willing to wait that long and have to return for treatment initiation. Costs of the Xpert assay may vary among high-income countries but will be unacceptably high for resource-poor settings. Some other commercial PCR-based POCTs or near-patient tests to detect *Ng* are now available. Truelab Realtime micro PCR system (Molbio) and Randox STI multiplex array that runs on the Bosch Vivalytic platform are both faster than GeneXpert and take only 50 and 30 min, respectively, but so far, evaluation of both tests has not been published in peer-reviewed papers [96]. The binx io platform (binx health, formerly Atlas Genetics) is based on PCR and electrochemical detection of amplified products. A duplex assay for *Ct* and *Ng* provides results in about 30 min and recently received FDA approval (Table 1). The assay was evaluated in a multicenter study with more than 1500 symptomatic and asymptomatic patients (ClinicalTrials.gov Identifier: NCT03071510). Performance data have not yet been published in a peer-reviewed paper, but according to the company, sensitivity and specificity for *Ng* were 100% and 99.9%, respectively (https://mybinxhealth.com/news/binx-health-receives-fda-510k-clearance-for-rapid-point-of-care-platform-for-womens-health/).

Other NAATs that use isothermal amplification, instead of thermal cycling as in PCR, may reduce both costs and the time to result. Horst et al. recently described a paperfluidic device, integrating swab sample lysis, nucleic acid extraction, DNA amplification by isothermal thermophilic helicase-dependent amplification, and visual detection of amplified products on lateral flow strips. Using this device, *Ng* has been detected with 95% sensitivity and 100% specificity in a proof-of-concept study on 40 urethral and vaginal swab swabs [97]. The turnaround time of this low-cost assay is 80 min and may, according to the authors, be reduced to 60 min, making the assay particularly useful in settings with limited resources. Using recombinase polymerase amplification (RPA) *Ct/Ng* infection can be detected even faster in approximately 15 minutes [98]. A prototype version of this assay (TwistDx RPA assay) runs on the battery-powered Alere i instrument (Alere, Waltham, MA, USA), independent of electricity from the socket. The preliminary evaluation of diagnostic accuracy to detect *Ct* and *Ng* showed sensitivity, specificity, PPV, and NPV > 94% for male FCU, female FCU, and self-collected vaginal swabs [98]. Several other novel developments combining NAAT with microfluidics and nanotechnology that link high sensitivity with rapidity are in the pipeline and represent promising strategies towards a reliable and inexpensive POC testing [96]. These assays are performed on small devices using various microfluidic platforms that direct fluids through channels and reaction chambers for sample preparation and target detection by mechanical and electrokinetic mechanisms. Due to the integration of analytical steps on miniaturized devices, they were also named lab-on-a-chip systems. For example, mCHIP is a portable microfluidic diagnostic device for the detection of antigens and antibodies [99]. It was arranged for diagnosing HIV and syphilis but may also be used for detection of *Ng* and *Ct* antigens. The Vivalytic Analyzer (Bosch) also uses microfluidic techniques for fully automated qualitative and quantitative PCR analysis. The instrument represents an open system able to process diagnostic tests from different manufacturers. Most NAAT-based rapid tests require specific instruments and current supply and are probably unaffordable for low-income countries. However, recently a low-cost, portable analyzer for NAAT-based POCT was described [100] that appears attractive for settings with limited resources.

## 7. Future Demands

Among the methods for *Ng* detection, NAATs are the most sensitive, but so far, the vast majority of commercially available NAATs does not provide any information about resistance against antimicrobial compounds. Due to the higher sensitivity and easier workflow, as well as more rapid, automated, and high throughput testing, laboratories are increasingly using NAATs instead of culture, leading to a reduction of phenotypic AMR data. In the absence of resistance data, patients were treated empirically, which in the past has led to development of resistance to virtually all antibiotics used for *Ng*, especially in case of monotherapy.

The emergence of AMR strongly impairs the efficacy of *Ng* treatment and represents a significant clinical and public health challenge. Thus, bacterial culture should be attempted whenever possible. When using NAAT as a primary diagnostic test, cultivation of *Ng* subsequent to positive NAAT results frequently fails due to the limited viability of *Ng*. Although deferred culture of the bacteria has been improved using novel flocked swabs for collection of clinical specimens, the probability of successful culture declines with increasing storage time at 4 ℃ (to 69% after one day and 56% after 2 days) [63], indicating the need of further improvement of specimen collection and culture methods. On the other hand, predictions of AMR might be derived from genotypic data obtained in the context of NAAT analysis, ideally using a rapid molecular POCT.

Molecular alterations causing resistance to antimicrobials used to treat *Ng* were well characterized [101,102], and a number of real-time PCR tests detecting resistance determinants have been published in the last years (see [102] for a review). The assays were designed to detect single resistance mutations or multiple mutations associated with resistance to particular drugs or to a class of antibiotic compounds, like quinolones, macrolides, or beta lactam antibiotics [103,104,105,106,107,108,109]. To our knowledge, there is currently only one commercial assay for genotypic resistance testing of *Ng* (SpeeDx ResistancePlus GC for quinolone susceptibility). Some other molecular techniques were published that use a MASSarray iPLEX platform, multiplex bead arrays, and multiplex PCR with high-resolution melting analysis or based on mismatch amplification, allowing a more comprehensive analysis of resistance to multiple antimicrobials [110,111,112,113].

Generally, these assays were shown to accurately differentiate between wild type and mutation in both clinical samples and isolated bacteria but differ with respect to performance characteristics. Clinical sensitivity and specificity of some assays is limited, mainly by low *Ng* load and cross-reactive species [102]. Furthermore, prediction of resistance to antimicrobial drugs based on detection of individual resistance determinants is difficult, as development of resistance requires accumulation of several determinants in most cases. In order to standardize interpretation of genetic alterations associated with antibiotic resistance, the Public Health Agency of Canada has developed a web-based system (NG-STAR) for classification of seven genes associated with resistance to three classes of antibiotics (cephalosporins, macrolides, and fluoroquinolones). Currently, prediction of resistance based on genetic changes is most accurate for fluoroquinolones [104,114,115]. The primary mutations associated with ciprofloxacin resistance are located in the quinolone resistance determining region (QRDR) of gyrA, encoding subunit A of DNA gyrase. Additional mutations in parC encoding a subunit of topoisomerase IV are required for high-level resistance [101]. In regions with less prevalent ciprofloxacin resistance (i.e., outside Asia) implementation of genotypic ciprofloxacin resistance testing appears useful, as shown in an American study describing a significant decline of ceftriaxone use when guiding treatment by a PCR-based assay for ciprofloxacin resistance [116].

However, resistance to other antimicrobials used in *Ng* treatment is more complex, as multiple mechanisms contribute to the development of resistance. For instance, resistance to beta lactam antibiotics includes expression of beta lactamases, altered PBPs (point mutation and mosaic variants), increased discharge by efflux pumps, and reduced uptake by porins [102]. Consequently, in these cases, the presence of individual AMR determinants is insufficient to predict phenotypic resistance, but on the other hand, exclusion of any resistance determinant may indicate susceptibility with high probability [104].

In addition to determining the level of resistance analogous to MIC values in bacterial culture, genotypic resistance testing of *Ng* faces several other challenges, like cross-reactivity, especially in extragenital samples [103,106]; mixed infections of susceptible and resistant strains [117]; and the necessity of internal controls to exclude false negative results due to low gonococcal concentrations [118]. In addition, genetic assays are able to detect only known but not new AMR determinants that frequently develop in *Ng*, as we have seen in the past. Thus, any putative commercial genotypic test needs to be modified accordingly in a reasonable period without time-consuming and costly clinical validation.

At present, application of genotypic resistance testing appears to be useful primarily for surveillance of gonococcal resistance. To guide individual therapy of patients, further improvements are required to achieve high diagnostic accuracy and a high predictive value of AMR. This may be achieved in the future by applying whole genome sequencing [47,119,120] and deep learning systems for evaluation of sequencing data that, based on large sets of correlated genotypic and phenotypic data, may provide quantitative information about resistance to particular drugs, similar to the algorithms used in guiding antiretroviral therapy, for instance.

## Figures and Tables

**Figure 1 pathogens-09-00091-f001:**
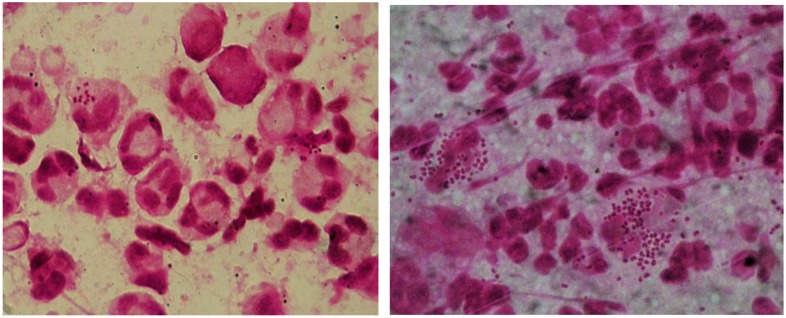
Gram stain from a male urethral swab, depiction of polymorphonuclear leukocytes (PMNL) with Gram-negative intracellular diplococci. The left and right picture represent two different microscopic slides prepared from the same swab.

**Figure 2 pathogens-09-00091-f002:**
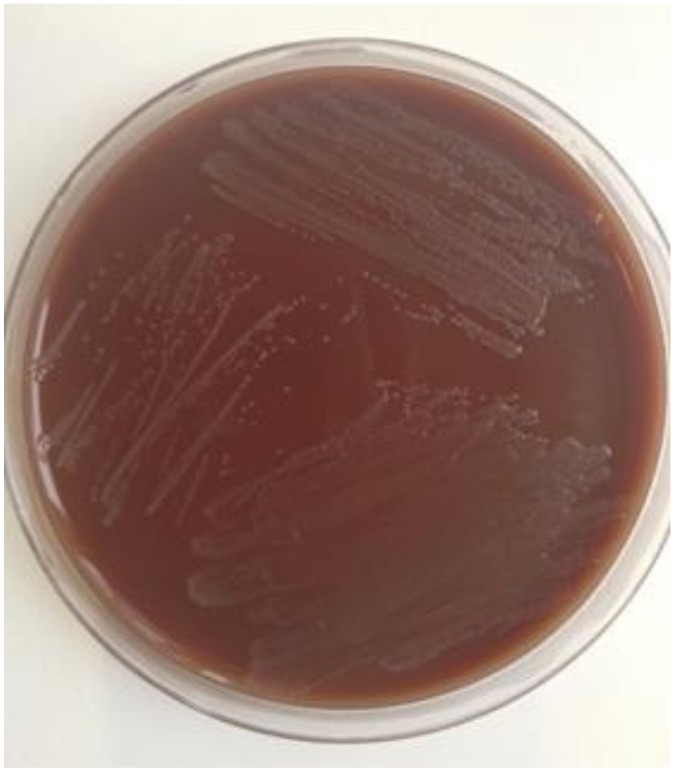
Culture of *Neisseria gonorrhoeae* (*Ng*) from a male urethral swab.

**Figure 3 pathogens-09-00091-f003:**
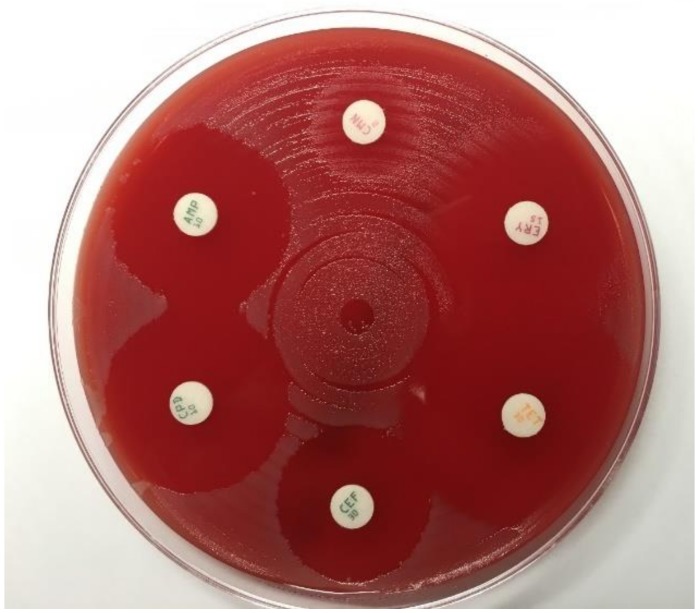
Disc diffusion assay (exemplary, no presentation of *Ng* testing). Photo by Andreas Gross (MVZ Laboratory Krone GbR, Bad Salzuflen, Germany).

**Figure 4 pathogens-09-00091-f004:**
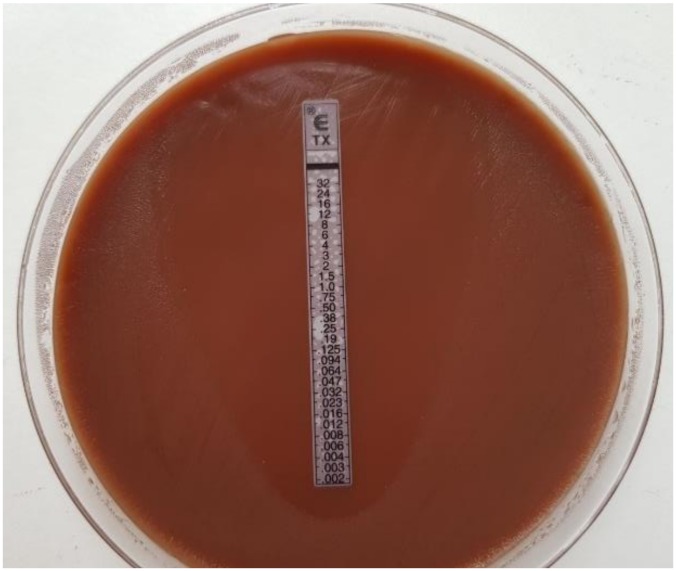
Minimum inhibitory concentration (MIC) gradient strip test method.

**Table 1 pathogens-09-00091-t001:** Current FDA-approved nucleic acid amplification technologies (NAATs) for detection of *Neisseria gonorrhoeae*.

Assay (Company)	*Ng* Targets	Cleared Specimen Types
Abbott RealTime CT/NG (Abbott)	Opa gene	Women: urine, swabs (vaginal, endocervical) Men: urine, urethral swab
cobas CT/NG (Roche)	Two different targets in the DR 9 region	Women: urine, swabs (vaginal, endocervical) Men: urine
APTIMA Combo 2 Assay (Hologic)	16S-rRNA	urine swabs (vaginal, endocervical, urethral, rectal, pharyngeal)
BD MAX GC BD MAX CT/GC BD MAX CT/GC/TV	OpcA gene	urine (20-60mL of first morning urine recommended), swabs (vaginal endocervical)
BD ProbeTec Neisseria gonorrhoeae (GC) Qx Amplified DNA Assay	Pilin-gene inverting protein homologue	Women: urine, swabs (vaginal, endocervical) Men: urine, urethral swab
BDProbeTec ET Chlamydia trachomatis and Neisseria gonorrhoeae Amplified DNA Assays	Pilin-gene inverting protein homologue	Women: urine, endocervical swab Men: urine, urethral swab
Xpert CT/NG (Cepheid)	Two distinct chromosomal targets	urine swabs (vaginal, endocervical, rectal, pharyngeal)
binx io CT/NG Assay (binx health)	Not specified	vaginal swabs

Notes: As of December 2019.

**Table 2 pathogens-09-00091-t002:** Multiplex PCR assays for STI testing.

Assay (Company)	Method of Amplification and Detection	Time to Result	Detected Pathogens
FTD STD 9 (FastTrack Diagnostics)	Real-time PCR Fluorescence	3–4 h	*Ct*, *Ng*, *Tv*, *Mg*, *Uu*, *Up*, *Gv*, HSV1, HSV2
Anyplex II STI-7 (Seegene)	Real-time PCR fluorescence and melting curve	4–5 h	*Ct*, *Ng*, *Tv*, *Mg*, *Uu*, *Up*, *Mh*
Amplisense (Interlab Service)	Real-time PCR Fluorescence	3–4 h	*Ct*, *Ng*, *Tv*, *Mg*
FilmArray STI (BioMerieux)	Nested PCR Fluorescence	1 h	*Ct*, *Ng*, *Tp*, *Tv*, *Mg*, HSV1, HSV2, *Uu*, *Hd*
Easy Screen (Genetic Signatures)	3-base real-time PCR (Bisufit-PCR) melting curve	3 h	*Ct*, LGV, *Ng*, *Mg*, *Tv*, *Uu*, *Up*, *Mh*, GBS, Candida, *Gv*, HSV 1, HSV 2, VZV, *Tp*
STI Multiplex Aray (Randox Laboratories)	Real-time PCR Fluorescence	30 min	*Ct*, *Ng*, *Mg*, *Tv*, *Uu*, *Mh*, *Hd*, *Tp*, HSV 1, HSV 2

*Ct: Chlamydia trachomatis, Ng: Neisseria gonorrhoeae, Tv: Trichomonas vaginalis, Mg: Mycoplasma genitalium, Uu: Ureaplasma urealyticum, Up: Ureaplasma parvum, Gv: Gardnerella vaginalis*, HSV: Herpes simplex virus, *Mh: Mycoplasma hominis, Hd: Hemophilus ducreyi*, LGV: Lymphogranuloma venereum, GBS: Group B streptococci, VZV: Varicella zoster virus, *Tp: Treponema pallidum*.

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
