# Peer review of "The Laboratory Diagnosis of Neisseria gonorrhoeae: Current Testing and Future Demands"

_pathogens, 2020, doi:10.3390/pathogens9020091_

Round 1

Reviewer 1 Report

This reveiw from Meyer and Buder examines the laboratory diagnosis of Neisseria gonorrhoeae, with emphasis on current testing and future demands. in essence, the authors have identified NAATs as the primary method(s) for identifying gonococci, as we expect, in the high-income setting. There is some discussion of point-of-care approaches, but these are still in their infancy. the review could be improved with the following suggestions/changes.

Line 30. The currnt estimate for Gc infections globally is around 87 million (Rowley, Bull WHO, 97 (8) 548 -). please update. Line 30. For the reader, the authors should give a breakdown of the global incidence of Gc infection. Of these 87 million, about 1 million are in the US, and perhaps similar numbers if Northern Europe/Russia. The BULK are in LMICs. perhaps a table to show the disproportionate load of infection between HICs and LMICs would make clear how important it is to have cheap POCs. I would like to see more discussion of clinical symtoms as diagnostic markers for STIs. I know that for uncomplicated gonrrhoea, discharge and pain are potential predictors, but are not specific. In the LMIC setting, clincial diagnoses are critical (line 48)

Line 71, a reference is needed for statment of non-Gc neisseria trasnferring resistance to Ng in the pharynges (indeed, might ven be non-neisseria transmission?)

Line 91-103. I presume pharyngeal sample microscopy could provide a confusing picture with the presence of other G-ve diplococci? Some pictures would provide variety for ther non-specialist reader.

Line 105. Bacterial culture is not the only method that allows AMR testing. There is some good evidence (as reported later in the review) that whole-genome sequencing is very useful. Some more info here on the method would seem appropriate (see Eyre et al., METH Mol Biol 2019).

Line 113- ; some pictures again of gonococci would be useful to have.

Line 127 – what are Gc serogroups? In Neisseria, we tend to think of serogrouping for meningococcal capsules.

Line 162. Again, some pictures of these assays would be useful – many non-specialists won’t know what these look like (agar diffusion, Etests, etc.)

Table 1, Table 2. It would be very important to have an idea of the actual costs for these tests. Indeed, the review lacks any indication of costs for testing for Gc using these assays. It’s important to know in the context of infection in LMICs.

POC assays – are they really going anywhere? The ones mentioned, i.e. PCR-based assays are not going to be cheap (again, how much?) and unlikely to be in use in Africa or the far east soon. I would like to see from lines 355- 357, more discussion of the microfluidic platforms the authors think might offer a solution for POC testing. This would be the novelty of the current review.

Adnex – not really necessary to have the method for gram’s stain – it’s such basic microbiology.

Reviewer 2 Report

In this review the authors review literature regarding methods to detect gonococcal infections and resistance to antibiotics. In the main, the review is timely and important, and generally well organized.  I have a few suggestions for the authors to consider:

Point of care diagnostics from detecting gonococci and AMR are critically needed. While the authors consider this there is some need to expand the section.  For instance, they should consider including the possibility of using a rapid phenotype-based system for detecting AMR through use of specific gonococcal gene expression during exposure to antibiotics. They should also include a bit more regarding limitations of POC diagnostics with respect to limits of detection, specificity/sensitivity, acceptable costs in the resource-poor regions of the world, etc. It would be helpful to the reader to include a table that summarizes specificity and sensitivity of diagnostics tests for detecting Neisseria gonorrhoeae in clinical specimens from genital and extra-genital sites. line 31: define MSM Please change NG to Ng. line 196: change "vital" to "viable". In all cases, genus and species should be italicized Line 241: give full genus name
